# Tristetraprolin Family Members and Processing Bodies: A Complex Regulatory Network Involved in Fatty Liver Disease, Viral Hepatitis and Hepatocellular Carcinoma

**DOI:** 10.3390/cancers17030348

**Published:** 2025-01-21

**Authors:** Noémie Gellée, Noémie Legrand, Mickaël Jouve, Pierre-Jean Devaux, Laurent Dubuquoy, Cyril Sobolewski

**Affiliations:** Univ Lille, Inserm, CHU Lille, U1286-INFINITE-Institute for Translational Research in Inflammation, F-59000 Lille, France; noemie.gellee@univ-lille.fr (N.G.); noemie.legrand@univ-lille.fr (N.L.); mickael.jouve@univ-lille.fr (M.J.); laurent.dubuquoy@inserm.fr (L.D.)

**Keywords:** tristetraprolin, MASLD, hepatocellular carcinoma, viral hepatitis, alcohol-related liver disease, processing bodies

## Abstract

Tristetraprolin (TTP) family members are major regulators of gene expression involved in metabolic, inflammatory and carcinogenic processes. These proteins can promote mRNA degradation through a complex regulatory process involving the assembly of membraneless cytoplasmic granules, namely, processing bodies (P-bodies). Alteration of the TTP/P-bodies axis contributes to the development of chronic liver diseases and their progression toward hepatocellular carcinoma (HCC). Herein, we discuss the current roles of TTP family members and P-bodies in the context of fatty liver disease, hepatic viral infections and HCC.

## 1. Introduction

Fatty liver disease (FLD) encompasses a spectrum of metabolic and histological alterations triggered by viral infections (hepatitis B and C viruses), chronic and abusive alcohol consumption (alcohol-related liver disease (ALD)) and obesity/type-2 diabetes (metabolic dysfunction–associated liver disease (MASLD)) [1,2,3,4]. The development of FLD starts with the accumulation of lipids in hepatocytes (steatosis), which, with time, progress toward chronic inflammation (steatohepatitis). Then, the accumulation of other types of damage, including oxidative stress, lipotoxicity, mitochondrial dysfunction and endoplasmic reticulum stress, triggers hepatocyte death and the activation of hepatic stellate cells (HSCs), which, in turn, foster hepatic fibrosis and the progressive loss of hepatic functions [3,5]. Due to the regenerative capacity of the liver, regenerative nodules appear in the parenchyma and further alter the liver architecture, and thus, its functions [3]. Cirrhosis is a major cause of mortality and a high-risk condition for the onset of hepatocellular carcinoma (HCC). HCC is currently the seventh most frequent cancer in the world, and the third most frequent cause of cancer mortality worldwide [4]. Due to the high prevalence of fatty liver diseases, an increase in the incidence of HCC is expected in the future [1,2]. Unfortunately, only few and poorly efficient therapeutic options are available for patients with hepatic fibrosis, cirrhosis and HCC, namely, surgery; liver transplantation [6]; or more recently, the thyroid hormone receptor-β agonist Resmetirom^®^ [7]. Deciphering the molecular mechanisms involved in the development of these disorders remains an urgent need for this major public health concern.

Epigenetic defects (e.g., DNA methylation, histones modifications) of gene expression importantly contribute to the alteration of several processes involved in FLD development (e.g., lipid/glucose metabolism, inflammation, fibrosis, liver regeneration, carcinogenic processes) [8,9]. Although tremendous efforts have been devoted to characterizing non-coding RNA functions, such as for microRNAs (miRNAs) or long-non-coding RNA (lncRNAs) [10,11], increasing evidence indicates that RNA-binding proteins (RBPs), such as AU-rich element-binding proteins (AUBPs), represent critical post-transcriptional regulators of gene expression in the liver [12]. Through their capacity to bind not only to the 3′UTR but also the 5′TL (transcript leader) or intronic regions, these proteins control mRNA stability, splicing and translation [13]. While some AUBPs stabilize mRNAs (e.g, Human antigen R (HuR)), others promote mRNA decay, such as tristetraprolin family members (i.e., tristetraprolin (TTP), Butyrate Response Factor 1 (BRF1) and Butyrate Response Factor 2 (BRF2)) [14]. Moreover, AUBPs can interfere with the miRNA-dependent regulation of gene expression [15]. TTP-mediated mRNA decay can occur in small cytoplasmic granules, commonly referred to as “Processing-Bodies” (P-bodies), which contain all the degradation machinery (decapping enzyme, deadenylase, etc.) [16]. In stress conditions, such as those encountered during metabolic-associated steatohepatitis (MASH) progression (e.g., lipotoxicity, endoplasmic reticulum (ER) stress and hypoxia), TTP localizes in other cytoplasmic compartments, namely, stress granules (SGs), where the mRNAs are kept translationally silent [13]. Alteration of TTP expression/activity but also of P-body/SG functions contribute to the development of chronic liver diseases and HCC. Herein, we review and discuss the different studies that dealt with the role of TTP family members and P-bodies in the development of FLD.

## 2. TTP-Dependent mRNA Regulation

### 2.1. AUBPs at a Glance

The post-transcriptional regulation of gene expression requires a variety of trans-acting factors able to control the processing, the stability and the translation of mRNAs [17]. While intense efforts have been devoted to characterizing non-coding RNAs, it is now clear that RNA-binding proteins are also key determinants of gene expression [15,18,19]. Among them, AU-rich element-binding proteins (AUBPs) are of particular interest due to their high affinity for AU-rich (ARE) motifs present in the 3′UTR of about 5–8% of the transcriptome [20]. These proteins also bind to ARE motifs in the 5′TL (transcript leader) and intronic regions [21], and thus, also play a role in mRNA splicing and translation [18,21,22]. Based on their sequence, ARE sequences have been divided into three different classes. The first class contains dispersed “AUUUA” motifs in the 3′UTR (e.g., c-Myc, *CCNA* (cyclin A), *CCNB1* (cyclin B1)). The second class consists of multiple ARE sequences (e.g., GS-CSF (Granulocyte colony-stimulating factor human), TNFα (Tumor Necrosis Factor α), COX-2 (cyclooxygenase 2)). Finally, the third contains repetitions of the U nucleotides (e.g., p53, c-Jun) [20]. AUBPs gather more than 20 proteins that are not only involved in mRNA stability but also mRNA splicing and translation [15,18]. Among them, the Elav (Embryonic Lethal Abnormal Vision) (e.g., HuR) and tristetraprolin family members (e.g., Tristetraprolin (TTP)) represent the most studied AUBPs due to their ability to control the expressions of key transcripts involved in metabolic (e.g., FGF21 (Fibroblast growth factor 21), PTEN (Phosphatase and TENsin homolog), PPARγ (Peroxisome proliferator-activated receptor gamma)), inflammatory (e.g., TNFα, COX-2) and cancer-related processes (e.g., c-Myc, VEGFα (Vascular Endothelial Growth Factor α)) [23]. In agreement, constitutive TTP KO mice develop a severe inflammatory syndrome [24]. Moreover, haplo-deficient TTPKO mice are more susceptible to colorectal carcinogenesis [24], thus highlighting the importance of AUBPs in cancer development.

### 2.2. TTP-Family Members: From Structure to Function

The TTP family belongs to the 2-O-tetradecanoylphorbol-13-acetate (TPA)-inducible sequence 11 (TIS11) family. The TTP family represents the Cysteine-Cysteine-Cysteine-Histidine (CCCH) zinc finger proteins gathering three members, namely, TTP (tristetraprolin, ZFP36 (Zing Finger Protein 36 homolog), TIS11), BRF1 (Butyrate response factor 1, ZFP36L1 (Zing Finger Protein C3H Type-Like 1), TIS11B) and BRF2 (Butyrate response factor 2, ZFP36L2 (Zing Finger Protein C3H Type-Like 2), TIS11D) [25]. This CCCH motif allows for the interaction with mRNAs, mainly in the 3′ UTR [26]. TTP, the most characterized, is considered as an immediate early response gene that can not only be rapidly induced by phorbol-myristate 13-acetate (PMA) but also various growth factors and cytokines (e.g., EGF (epidermal growth factor), IGF-1 (insulin-like growth factor 1), TGF-β (Tumor Growth Factor β), insulin, glucocorticoids) [27,28,29,30,31,32]. TTP contains two zinc finger motifs flanked by intrinsically disordered regions (IDRs) [33] and surrounded by three quadruple proline (PPPP) motifs, which are required for the interaction with the translational repressor complex 4EHP-GYF-2 (Eukaryotic Translation Initiator Factor 4E Nuclear Import Factor 1) [34] (Figure 1A). Finally, TTP contains a nuclear localization and export signal (NLS and NES), as well as a NOT1-binding domain required for the binding of TTP with the degradation machinery (CCR4/NOT complex) [35,36]. More precisely, this domain allows for the interaction with CNOT1 (CCR4-NOT Transcription Complex Subunit 1) of the CCR4/NOT complex [26,35,37] (Figure 1B). Then, through the recruitment of the deadenylase CAF1 (CCR4-associated factor 1, also called CNOT7), this complex promotes the deadenylation of mRNAs [36,37]. This represents the first step for mRNA degradation mediated by TTP and P-body formation.

### 2.3. TTP-Family Members: A Complex Regulatory Mechanism?

Similar to miRNAs, the regulation of gene expression by AUBPs is complex since one AUBP can control a myriad of mRNAs and, conversely, one mRNA can be targeted by several AUBPs [15]. Bioinformatic tools are available and allow for an easy prediction of AUBPs targets based on the presence of ARE in the 3′UTR or intronic regions (e.g., ARE site, ARED database), or their binding to mRNAs (CLIP databases, e.g., POSTAR database) [38,39]. The capacity of AUBPs to interfere with miRNA-/long-non-coding RNA-dependent regulation adds another layer of complexity and raises important questions regarding the measurement of miRNA expression rather than their activity in human diseases [15]. These findings suggest that the post-transcriptional regulation of gene expression is much more complex than originally thought, and it is likely that the development of human diseases results from a concerted alteration of a vast network of RBPs and non-coding RNAs rather than a single trans-acting factor. Although this complexity has been extensively documented for HuR [15], only sparse information is available for TTP family members. TTP competes with HuR for binding to several transcripts (e.g., TNFα, VEGFα) [15]. Therefore, not only the development of chronic inflammation but also the overexpression of oncogenes in cancer may result from a double hit, with the loss of TTP expression on one hand, and the upregulation of HuR on the other hand. Interestingly, TTP can also cooperate with miRNAs, such as let-7, to downregulate specific oncogenes (e.g., CDC34), as evidenced in colorectal cancer [15,18]. Such an interplay with miRNAs is even less characterized for the other TTP family members but deserves a thorough investigation for a better characterization of human diseases. Moreover, it is currently unknown whether TTP family members interfere with lncRNAs. Herein, we discuss not only the role of TTP family members but also of P-bodies in the development of FLD and HCC.

### 2.4. TTP-Mediated mRNA Regulation

Although, most studies depicted the role of TTP in mRNA decay, it is clear now that depending not only on its localization but also on post-translational modifications, TTP exerts other important functions in RNA metabolism (i.e., mRNA translation and splicing).

*TTP and P-body-mediated mRNA decay:* P-bodies are small cytoplasmic ribonucleoproteins granules composed of translationally repressed mRNAs and proteins involved in mRNA decay [16] (Figure 2). The degradation machinery gathers decapping enzymes (e.g., DCP1A (mRNA decapping enzyme 1A), EDC3 (enhancer of mRNA decapping protein 3), EDC4 (enhancer of mRNA decapping protein 4), the 5-3′ exonuclease XRN1 and the CCR4/NOT (Carbon catabolite repression 4- negative on TATA less) complex) that promote the deadenylation of mRNAs [16]. Although P-bodies are present in physiological conditions, their assembly is triggered by stressful stimuli (e.g., infections, oxidative stress) [16,40,41]. The mechanism involved in their formation is still poorly understood but requires a liquid–liquid phase separation (LLPS), which is promoted by weak interactions between proteins bearing an intrinsically disordered domain (e.g., TTP, EDC4, DDX6 (Dead box helicase 6), Lsm (Sm-like protein) 14A or 4E-T) and/or a RNA-binding domain, as well as untranslated mRNAs [16,42]. Therefore, P-bodies are membraneless compartments exhibiting liquid-like properties and allowing the rapid exchange of proteins and mRNAs with the cytosol [16,40,42]. The RNA degradation is a sequential process, starting with the deadenylation of mRNAs by the PAN2/3 (PolyA specific Ribonuclease ribonuclease) complex, which recognizes the PolyA-Binding Proteins (PABPs) bound to the polyA tail [43]. When the tail is smaller than 110 nucleotides, the CCR4/NOT and the deadenylase CAF1 (CNOT7) [43] mediates the deadenylation process [43]. CCR4 removes the PABP from the RNA tail, and CAF1 degrades the free polyA tail. Then, the free 3′ RNA tail is targeted by the TUT4/7 (terminal uridylyltransferases 4/7) complex, which adds a small U tail, a marker of deadenylated RNA. This U tail is a target for Lsm1-7 complexes and allows for mRNA degradation. Finally, the decapping enzyme DCP2 (Decapping mRNA 2)/DCP1A removes the 5′ cap and exoribonucleases, such as the 5′ to 3′ exonuclease XRN1, and degrades the mRNA [16,40,43].

Although P-bodies are mostly involved in mRNA decay, increasing evidence suggests that they are also involved in RNA storage [16,40]. P-bodies are key components of mRNA metabolism, and thus, the alteration of their assembly contributes to the development of several diseases, including cancers. In cancer cells, increasing evidence indicates that P-bodies promote cell proliferation and migration [41]. However, their roles in HCC and chronic liver diseases remain poorly documented and controversial.

*TTP-mediated mRNA translation inhibition:* The mechanism of TTP-mediated translation repression is still poorly known. Nonetheless, TTP interacts with the cap-binding translation repression complex 4EHP (also named eIF4E2 (Eukaryotic translation initiation factor 4E type 2))-GYF2 [44] through tetraprolin motifs, thereby preventing the binding of other translation initiation factors (e.g., eIF4G, eIF4A, eIF4B) [34,45] (Figure 2). Finally, TTP may also repress mRNA translation through the recruitment and sequestration of mRNAs into stress granules (SGs) (Figure 2). Similar to P-bodies, these cytoplasmic compartments are formed in harmful conditions in an LLPS-dependent manner [40,42]. In contrast to P-bodies, it is currently thought that SGs do not degrade RNA but keep them translationally silent [40,46].

*Other TTP-mediated effects:* Besides its canonical function in mRNA decay and translation, TTP is also involved in mRNA splicing, as evidenced in HeLa cells, where TTP regulate the alternative splicing of some genes involved in the immune response and NF-κB (Nuclear Kactor κ B) pathway, such as *CXCL2* (Chemiokine (C-X-C motif) ligand 2)*, IL1R1* (Interleukin 1 receptor type 1) *or RELB* (V-Rel Avian Reticuloendotheliosis Viral Oncogene Homolog B (Nuclear Factor Of Kappa Light Polypeptide Gene Enhancer In B-Cells 3)) [47]. Moreover, TTP interacts with the polyA-binding protein nuclear 1 (PABPN1), leading to an inhibition of the polyadenylation of mRNA targets (e.g., IL-10 or TNFα). This interaction between TTP and PABPN1 is decreased by TTP phosphorylation by MK2 (MAP kinase activated protein kinase 2). In the nucleus, TTP decreases the transcriptional activation of NF-κB and prevents p65 (nuclear factor-kappa β) nuclear translocation in HeLa and human embryonic kidney cell lines and in mouse embryo fibroblasts [48]. Although NF-κB activation contributes to hepatic inflammation, fibrosis and HCC [49], the role of TTP on NF-κB activation in the context of the liver was never investigated. Finally, TTP can also stabilize mRNA, as evidenced in iNOS (human inducible nitric oxide synthase) in the human adenocarcinoma cell line. In this case, TTP binds and inhibits the KSRP (KH-type splicing regulatory protein)-mediated mRNA decay of iNOS [50,51]. This interplay between TTP and KSRP challenges the dogmatic view of TTP as a strict mRNA decay factor.

## 3. TTP Family Members Regulation in the Liver

The regulation of TTP family members occurs at different levels (i.e., transcriptional, post-transcriptional and post-translational) and the alteration of the involved mechanisms strongly contributes to the onset of numerous diseases and cancers. Furthermore, the environment plays a critical role in both the expression and activity of TTP family members. In a normal liver, *ZFP36L1* is widely expressed, while *ZFP36* and *ZFP36L2* are poorly expressed in hepatocytes, as evidenced in single-cell RNAseq analyses (Figure 3). However, the alterations of their expression/activity have been associated with the development of several liver diseases and HCC.

### 3.1. TTP

The expression of TTP is regulated at the transcriptional level by different signaling pathways. First, TTP expression is increased by pro-inflammatory stimuli (e.g., IL-1β, TNFα), which trigger the NF-κB pathway, thereby promoting *ZFP36* transcription [13,30]. The IL-4/STAT6 (Signal Transducers and Activators of Transcription 6) induces TTP expression in mast cells, and thus, decreases the TNFα production [13]. Besides this inflammatory signaling, the TTP level is also stimulated by insulin signaling in adipocytes, thus highlighting its metabolic role [52,53] (Figure 4). *ZFP36* expression is also induced by other transcription factors, including Activating-Protein 1 (AP-1) [27,28,29], the glucocorticoid receptor (GR) and Kruppel-like factor 4 (KLF4) [31]. Similarly, the oncogene MYC inhibits *ZFP36* transcription in lymphoma [54]. In contrast, the tumor suppressor p53 directly binds to the *ZFP36* promotor and triggers its transcription, as evidenced in ovarian cancer cells [55]. Alterations of the expression of transcription factors contribute to the loss of TTP in many cancers. This is the case for Early growth Response 1 (EGR1), a major transcription factor involved in *ZFP36* transcription, which is downregulated in several cancers, including colorectal cancer or HCC [18,30,56,57].

At the post-transcriptional level, TTP is regulated by a wide range of miRNAs (e.g., miR-29a), whose expression is upregulated in some cancers (e.g., miR-29a in melanoma). In the liver, a few miRNAs regulating TTP have been described, such as miR-182-5p in the context of parasitic infection by *Schistosoma japonicum* [58]. This regulation is probably underestimated given that several miRNAs able to directly target TTP mRNA are strongly deregulated in liver diseases, such as miR-17-3p or miR-1207-5p in HCC [59,60] or miR-27 and miR-29a in hepatic fibrosis [61,62] (Figure 3). Future works aimed at characterizing these links may unravel potential therapeutic approaches.

At the post-translational level, several phosphorylation sites of TTP were identified and are mediated by various kinases, such as PKB (protein kinase B, AKT), ERK (Extracellular signal Regulated Kinase), GSK3β (Glycogen synthase 3 β), PKC (Protein Kinase C), JNK (c-Jun N-terminal Kinase) or p38 MAPK (Mitogen Activated Protein Kinase) [23,63,64,65]. While their roles remain poorly known, some are well-characterized determinants of TTP activity and localization. Among them, the p38 MAPK/MK2 plays a major role in TTP regulation. Indeed, the phosphorylation of TTP on Ser60 and Ser186 in human, or Ser52 and Ser178 in mouse by MK2, triggers TTP sequestration by the 14-3-3 chaperone in the cytosol [26,66,67] (Figure 3). Moreover, this phosphorylation by MK2 inhibits the recruitment of CAF1, and thus, impairs mRNA deadenylation in a 14-3-3 independent manner [68]. In contrast, the phosphatases DUSP1 (Dual-specificity phosphatase 1) or PP2A (protein phosphatase 2A) inhibit the MK2-dependent inactivation of TTP by 14-3-3 interaction [23,64,69,70,71]. Moreover, under LPS (lipopolysaccharides) stimulation, the phosphorylation of TTP at Ser316 by MK2 or RSK1 (p90 ribosomal S6 Kinase 1) prevents TTP interaction with the CCR4-NOT complex in macrophages, thereby impairing the mRNA destabilizing activity of TTP (e.g., on TNFα) [35]. During liver injury, the P38/MK2 axis is activated and contributes to the upregulation of immediate response genes (i.e., COX-2, c-MYC) [72]. MK2 depletion strongly attenuates these inductions in a TTP-dependent manner [72]. In HCC cells, MK2 inhibition promotes TTP function, thus reducing the expression of target genes (i.e., c-MYC, IER3 (Immediate Early Response 3), AKT-1) and triggering apoptosis [73].

### 3.2. BRF1 and BRF2

The regulation of BRF1 and BRF2 is poorly documented and totally unknown in the liver. At the transcriptional level, *ZFP36L1* and *ZFP36L2* are repressed by DNA methylation in glioma and myelofibrosis [74,75]. Post-translational regulations (e.g., phosphorylation) have also been described for BRF1 and BRF2 regulations. For instance, in hepatic cancer cells (i.e., HepG2), the AKT inhibitor triciribine triggers the ERK/RSK1 pathway, which phosphorylates BRF1 and reduces its binding to CNOT7, thereby increasing the stability of low-density lipoprotein receptor mRNAs [76]. Environmental factors are also important determinants of BRF1 and BRF2 regulation, as evidenced in mouse macrophages (RAW 264.7 cells), where LPS decreases both the *ZFP36L1* and *ZFP36L2* mRNAs while inducing *ZFP36* expression [77]. Interestingly, the downregulation of *ZFP36L1* and *ZFP36L2* is mediated by TTP. Moreover, LPS promotes BRF1 phosphorylation on serines 92 and 203, and this triggers its interaction with 14-3-3, thereby inhibiting its mRNA decay activity [77]. Finally, several miRNAs are involved in *ZFP36L1* and *ZFP36L2* regulations and are deregulated in hepatic disorders (e.g., mir-96 or miR-124-3p for *ZFP36L1*; miR-409-3p, miR-375 or miR-429 for *ZFP36L2*) (Table 1).

## 4. Role of TTP Family Members and P-Bodies in Viral Hepatitis

Viral hepatitis is triggered by different infectious agents, ranging from the A to E hepatitis viruses [115]. These infections can cause both acute and chronic liver damage. Although the spectrum of liver alterations can differ between viruses and genotypes, some can trigger hepatic steatosis, followed by steatohepatitis, cirrhosis and HCC [116]. The infection by hepatitis virus can occur not only via contaminated fluids (i.e., HAV (hepatitis A virus), HBV (hepatitis B virus), HCV (hepatitis C virus)) but also by a fecal–oral route from water and food (i.e., HAV) or by sexual intercourse (B, C and D) [115]. Although efficient vaccines and direct anti-viral agents (HCV) have been developed, these treatments are still poorly available in some countries, including those in Asia, Africa and South America, where hepatic viral infection are endemic [117]. While the major risk factors of hepatic cirrhosis and HCC in developed countries are abusive alcohol consumption and obesity/type 2 diabetes, HBV and HCV remain the first risk factors in Asia and Africa [118]. Currently, the roles of TTP family members and of P-bodies in viral hepatitis are poorly known and are only restricted to HBV and HCV.

### 4.1. Hepatitis B Virus

HBV infection represents the main cause of viral hepatitis, which affects almost 300 million people worldwide [119]. About 30% of chronically HBV-infected patients develop cirrhosis after 10 years and 20% will evolve toward HCC [120]. In 2017, 29% of cirrhosis-related deaths were linked to HBV infection [121]. These viruses enter host cells by endocytosis, and the uncoating of the viral genetic information occurs in the cytoplasm. Using the host machinery, viral proteins are produced, and the nucleic acids are replicated [122]. HBV infection increases the risk of cancer development, including HCC [123], pancreatic cancer [124] and diffuse large B cell lymphoma (DLBCL), a heterogeneous aggressive malignancy caused by B cells [125]. The role of TTP in the context of hepatitis B viral infection is poorly known. To date, one study documented a downregulation of TTP in the CD4+ and CD8+ T lymphocytes, CD14+ monocytes and PBMCs (peripheral blood mononuclear cells) of HBV-infected patients [126]. In agreement with the anti-inflammatory function of TTP, its knockdown in PBMCs from healthy donors promotes a strong increase in the expression of pro-inflammatory cytokines (e.g., IL-1β, IL-6, IL-8, IL-13), which are normally increased in the plasma of HBV patients [126]. Mechanistically, the downregulation of TTP in HBV patients is mediated by IL-8 and RANTES, which are secreted by PBMCs upon HBsAg stimulation, in a PKC (Protein Kinase C)-dependent manner [126]. Finally, the downregulation of TTP has been correlated with the prognosis and recurrence of HBV-induced HCC [127].

### 4.2. Hepatitis C Virus

HCV infection represents a high-risk factor for developing chronic liver diseases [118,128], and it is estimated that 2 to 3% of the population are infected by this virus [118,129]. The HCV genome contains an open reading frame (ORF) encoding structural (core, E1, E2 and p7) and nonstructural (NS2, NS3, NS4A, NS4B, NS5A and NS5B) proteins flanked by 5′ and 3′ UTRs. Of note, the 5′ UTR of the HCV RNA contains two specific binding sites for the liver-specific miR-122, an essential miRNA for HCV mRNA stability and HCV replication [130,131,132].

Currently, the role of TTP family members in HCV is unknown. Nevertheless, increasing evidence indicates that several P-body-related proteins are involved in HCV translation and/or replication [133,134,135,136,137,138,139]. The P-body proteins, XRN1 (exonuclease) and MOV10 (Moloney leukemia virus 10) inhibit HCV amplification and promote HCV degradation [140], thus suggesting that P-bodies are protective against HCV. In agreement, miR-122 protects the HCV genome from XRN1-mediated HCV degradation [141], while other P-body components, such as Ago-2 or Lsm1, cooperate with miR-122 to induce HCV translation and/or replication [138,142]. Finally, a recent study demonstrated that DDX6 promotes the binding of miR-122 to the 5′ UTR of HCV to ensure its translation [143].

During HCV infection, the disruption of P-bodies and a delocalization of some components, such as PatL1 (Pat 1 Like 1), Lsm1–7 (Sm-like protein 1–7), Ago-2 (Argonaute RISC Catalytic Component 2) and DDX6, occur at the site of HCV replication [133,134,135,136,144]. Although the site of delocalization of P-body proteins is still under investigation, recent findings suggest that in the JFH1 HCV clone, PatL1, Lsm1–7, Ago-2 and DDX6 are delocalized with HCV core protein around lipids droplets to ensure virus translation and replication [133,145,146]. However, with the Jc1-HCV strain (genotype 2a), a colocalization of Lsm1 and the core HCV protein was observed in the endoplasmic reticulum [134], thus indicating different mechanisms depending on the strain. Since the site of HCV assembly is around lipid droplets [147], the redistribution of the P-body proteins necessary for HCV replication seems to occur at this location. In agreement, the P-body component DDX3 co-localizes with the core protein and DDX6 in lipid droplets and promotes HCV replication [132,137,148,149,150,151]. Liver biopsies of patients demonstrate a decrease in the number of P-body-containing DDX6 and DCP1 in hepatocytes due to a re-localization of these proteins in HCV-infected patients [152]. Accordingly, the P-body formation was restored upon viral infection reduction [152]. Although PatL1, Lsm1–7 and DDX6 redistribution induce a change in the P-body composition [134], as well as a decrease in the number of P-bodies [145], the level of expression of these proteins remains unchanged [145] (Figure 4).

## 5. Role of TTP Family Members in Alcohol-Related Liver Diseases

Alcohol-related liver disease (ALD) is a major public health concern [153] and is caused by chronic and excessive alcohol consumption. ALD is responsible for 80% of liver-related deaths in Europe [154]. As for MASLD, ALD is associated with a spectrum of liver alterations starting with hepatic steatosis and progressing toward alcoholic steatohepatitis and cirrhosis [155]. Moreover, in some patients, acute hepatitis (AH) can occur and is characterized by severe liver failure associated with high short-term mortality [156].

Although there are a growing number of studies on ALD, the precise role of TTP family members in this context remains largely unknown. However, it was shown that alcohol increases the permeability of the intestinal epithelium for endotoxins and LPS, which rapidly reach the liver via the portal vein [157]. LPS activates the MAPK family members, such as the p38 and c-jun-N-terminal kinase (JNK), resulting in the production of pro-inflammatory cytokines (e.g., TNFα) [158]. Although the activation of the p38 MAPK by LPS has been associated with an increase in TNFα mRNA stability by inhibiting TTP [159,160,161], this link has not been demonstrated yet in the context of ALD. To date, only one study depicted the role of ZFP36L1 in ALD. Liver-specific Zfp36l1KO mice showed protection against the development of steatosis after a Lieber–DeCarli liquid diet, as well as an increase in *FGF21* mRNA, a protective factor of steatosis [162], which is directly inhibited by BRF1 [163] (Figure 5A).

Other factors are impacted during ALD, such as miRs [10]. Some of these miRs are known to regulate TTP expression, as mentioned in Table 1. Examples of *ZFP36* regulators include miR-29a, a potential biomarker for ASH [164]; miR-182, an inflammation promoter [165,166]; and miR-129-5p, a fibrosis regulator [167] for *ZFP36L1*. Bioinformatic analysis of the expression of TTP family members in patients with alcohol-related cirrhosis or alcohol-related hepatitis reveals a significant decrease in the LSM7 gene, as well as a significant increase in ZFP36 and ZFP36L2 in cirrhosis and Lsm4 in alcoholic hepatitis (Figure 5B,C). The role of P-bodies in ALD has not been characterized yet. Given the importance of TTP and P-bodies in other chronic disorders and cancers, these results deserve further investigations and may offer new therapeutic avenues and/or biomarkers.

## 6. Role of ZFP36 Family Members and P-Bodies in MASLD

### 6.1. TTP Family Members in MASLD

MASLD starts with the accumulation of neutral lipids (i.e., triglycerides, cholesterol esters) in hepatocytes, a step commonly referred to as “steatosis”. With time, lipotoxicity and oxidative stress trigger inflammation and hepatocytes to cell death, thereby progressing toward fibrosis [168]. TTP plays an important role in the pathogenesis of MASLD through its ability to control hepatic glucose and lipid metabolism. Indeed, liver-specific and total TTP KO are protected from hepatic steatosis [30,57,169] (Figure 6). Interestingly, as previously demonstrated in the context of alcohol for ZFP36L1 [163], this effect is tightly associated with the ability of TTP to trigger the decay of *FGF21* mRNA (Figure 6), a protective hepatokine, which reduces fat accumulation in hepatocytes and improves insulin sensitivity. Moreover, TTP has been highlighted as an important promoter of *HMGR* (3-Hydroxy-3-methylglutaryl coenzyme A reductase) transcription, which, in turn, promotes cholesterol biosynthesis [170]. Conversely, the AUBP HuR, which competes with TTP for the binding to several mRNA targets, exerts an opposite function in MASLD, as evidenced in liver-specific HuR KO mice, which develop hepatic steatosis and fibrosis [171,172]. This phenotype is not only associated with a downregulation of ApoB (apolipoprotein B) [172] but also of CYCS (cytochrome c), NDUFB6 (NADH deshydrogenase ubiquinone 1 beta subcomplex submit 6) and UQCRB (ubiquinol-cytochrome c reductase binding protein). Together, these findings suggest that MASLD development is associated with a dynamic interplay between TTP and HuR. The protective effect of TTP loss on hepatic lipid accumulation was further highlighted in total TTP KO mice fed a high-fat diet (HFD) or a methionine–choline deficient diet (MCD) [169]. However, lipid accumulation occurs in isolated primary hepatocytes isolated from TTPKO mice treated with oleate, thus suggesting that other hepatic cell types hinder lipid accumulation in vivo. Moreover, in this study, metformin was unable to reduce lipid accumulation in primary hepatocytes from TTPKO mice, thus suggesting that TTP is required for the beneficial effect of metformin in hepatocytes [169]. Co-culture experiments demonstrated that this contrasting phenotype is due to an interplay between Kupffer cells and hepatocytes. Besides its role in lipid metabolism, TTP expression is upregulated in a high-glucose condition and promotes gluconeogenesis due to its capacity to directly bind and degrade *SIK1* (Salt-induced Kinase 1) mRNA, as evidenced in hepatic cancer cells (i.e., HepG2) [173]. Regarding the other TTP family members (i.e., BRF1 and BRF2), only one study has demonstrated that mice depleted for *ZFP36L1* are protected from diet-induced obesity and steatosis via an impaired lipid absorption regulation of Cyp7a1 (cholesterol 7α-hydroxylase) and the bile acid level [174]. Interestingly, the activation of the nuclear receptor FXR (Farnesoid X Receptor) induces ZFP36L1 expression, which, in turn, directly binds and decreases the expression of Cyp7a1, thereby shutting down the bile acid synthesis [174].

### 6.2. P-Bodies in MASLD

The role of P-bodies in MASLD is poorly documented. However, similar to TTP, the CCR4-NOT complex, and more specifically the CNOT6L subunit, is an important regulator of hepatic metabolism [175,176,177]. The loss of CNOT6L in mice protects against diet-induced obesity and improves insulin sensitivity. As described before for TTP, this effect is due to the upregulation of FGF21, thus suggesting that a TTP/P-bodies axis regulates hepatic metabolism and insulin sensitivity [175]. Other subunits of the CCR4-NOT complex have been associated with hepatic lipid metabolism. The subunit CNOT7 (CAF1) is increased in the hepatic cell line model of steatosis [178], and the deletion of CNOT7 is associated with a decrease in steatosis and lipogenic genes, including *Fasn* (Fatty acid synthase), *Acaca* (Acetyl-CoA Carboxylase alpha Protein), *Scd1* (Stearoyl-CoA desaturase 1), *Srebf1* (sterol regulatory element-binding protein 1) and *Pparg*. Moreover, the absence of CNOT7 promotes the expression of fatty-acid-oxidation-related genes, including *Pdk4* (Pyruvate Dehydrogenase Kinase 4) and *Ppara* [178]. Heterozygous CNOT3 +/- mice are leaner than control littermates and are protected from lipid accumulation in the liver and adipose tissue when fed an HFD [179]. The downregulation of CNOT3 markedly improves glucose tolerance and insulin sensitivity [179]. Similar results were obtained in *ob/ob Cnot3+/−* mice. In the liver, this phenotype is associated with an increase in fatty acid oxidation genes (e.g., *Pdk4*, *Cpt1b* (Carnitine Palmitoyltransferase 1B), *Hsd17b6* (Hydroxysteroid 17 Beta Dehydrogenase 6), while lipogenic genes (e.g., *Elovl6*, *Scd1*, *Srebf1*) are downregulated [179]. Finally, liver-specific CNOT1 deficiency triggers lethal hepatitis associated with hepatocytes necrosis [37]. Moreover, a decrease in lipid metabolism genes (e.g., *Acly* (ATP citrate lyase), *Scd1*, *Acox1* (Acyl-CoA oxidase 1)) and an increase in cell cycle/DNA-damage-response genes (e.g., *Brca1* (Breast Cancer 1), *Bax* (Bcl-2-associated X protein)) were observed [37].

Collectively, P-body components seem to promote hepatocyte survival, hepatic steatosis and insulin resistance. However, a recent study challenged this hypothesis. Indeed, this study suggested that in the liver, P-body assembly is regulated by TRIM24 (Tripartite motif containing 24). This protein is phosphorylated on serine 1043 by the insulin pathway, leading to its translocation from the nucleus to the cytoplasm of hepatocytes. Once phosphorylated, TRIM24 interacts with EDC4 and triggers its ubiquitination, thereby decreasing its activity. This effect increases the stability of *Pparg* mRNA, which, in turn, promotes hepatic steatosis [180]. Therefore, this study suggested that P-bodies may protect against steatosis. Nevertheless, it is unclear in this study whether the observed phenotype in vivo is strictly dependent on the TRIM24/P-body axis. Indeed, the loss of TRIM24 in mice has been associated with the development of hepatic steatosis, fibrosis and hepatocellular carcinoma [181,182]. Altogether, these findings suggest that P-body components are key regulators of hepatic metabolism and insulin sensitivity, and thus, the alteration of their expression/activity may considerably contribute to the onset of MASLD. Nevertheless, it remains unclear whether these effects are mediated in a P-body-dependent or P-body-independent manner. Further investigations are required to better delineate the role of P-bodies in MASLD.

TTP/P-bodies in inflammation and fibrosis: TTP is a well-known negative regulator of inflammation, thanks to its capacity to directly promote the degradation of inflammatory-related transcripts (e.g., TNFα, COX-2) [183]. Inflammation is an important component of fatty liver disease, which contributes, together with hepatocytes cell death, to the development of hepatic fibrosis. Therefore, the development of these stages requires the interplay between hepatocytes, immunes cells (e.g., Kupffer cells (KCs)) and hepatic stellate cells (HSCs). Interestingly, a co-culture experiment revealed that the loss of TTP in murine KCs triggers hepatocyte cell death by necroptosis in a TNFα/MLKL (mixed lineage kinase domain-like pseudokinase)-dependent manner pathway [169] (Figure 6). In HSCs, the role of TTP is still controversial. TTP expression is reduced in activated HSCs by the lncRNA (long-non-coding RNA) linc-SGR1 (*Schizaphis graminum* resistance) and an increase in linc-SGR1 expression was observed in patients with hepatic fibrosis. The overexpression of TTP in HSCs (LX2 cells) prevents TGFβ-induced HSC activation by decreasing the linc-SGR1 expression [184]. While this study suggests that TTP inhibits HSCs activation, another study demonstrated that TTP is also an important inhibitor of ferroptosis in HSCs by destabilizing autophagy-related transcripts (e.g., autophagy related 16 like 1 (ATG16L1)). Sorafenib and erastin, two ferroptosis inducers, markedly reduce hepatic fibrosis in mice subjected to a bile duct ligation and the overexpression of TTP in HSCs (through vitamin A-coupled liposomes bearing *Zfp36* plasmid) strongly reduces this beneficial effect. Finally, the expression of TTP is strongly reduced in primary HSCs from sorafenib-treated patients [185].

Altogether, these findings clearly indicate that TTP is a key player in MASLD/MASH development. Further investigations are still required to not only fully characterize this protein but also the other TTP family members (i.e., BRF1, BRF2).

## 7. Role of ZFP36 Family Members and P-Bodies in HCC

### 7.1. TTP and HCC

The expression of TTP is downregulated in HCC [30]. Different mechanisms were proposed, including the methylation of a single CpG island within the *ZFP36* promoter [186] or the loss of the transcription factor EGR1 in a HDAC (Histone Deacetylase)-dependent manner [18,30,56,187]. Targeting these silencing mechanisms may represent an efficient strategy to restore TTP expression in cancer cells. Accordingly, the treatment of HCC cell lines with a DNA-demethylating agent (5-aza dC) and an MK2 inhibitor restores the TTP expression and activity in HCC cells, thereby decreasing the expression of TTP targets (e.g., *c-MYC*, *IER3* or *AKT*) and inducing cell death [73]. Similarly, the expression of TTP can easily be restored by treating HCC cells with HDAC inhibitors (e.g., trichostatin-A, SAHA, sodium butyrate). Interestingly, in our previous study, we found that the loss of TTP mostly occurred in high-grade, poorly differentiated HCC. These findings are compatible with in vitro findings showing a tumor-suppressive function of TTP in hepatic cancer cells [30,187,188]. Indeed, the depletion of TTP promotes hepatic cancer cell line migration, proliferation by an increased c-Myc expression [186] and invasion, while TTP overexpression induces apoptosis [188]. TTP is also a tumor suppressor, which directly targets *HIF1A* (hypoxia inducible factor 1 subunit α) mRNA, a protein involved in hypoxia and tumoral resistance [189]. Although, these in vitro data are consistent with the established tumor-suppressive function of TTP [27,56], these results were challenged by in vivo findings. Indeed, a first study, where liver-specific TTPKO mice were treated with the hepatic carcinogen diethylnitrosamine (DEN), demonstrated that the loss of TTP, specifically in hepatocytes, reduces the hepatic tumor burden [30,188]. This decrease was associated with a decreased monocyte infiltration in the liver. Similar findings were obtained in our own study, and we correlated this phenotype with the overexpression of FGF21, which is a direct TTP target and a potent tumor suppressor in the liver [190,191] (Figure 7).

### 7.2. BRF1 and BRF2 in HCC

In an HCC cell line, the downregulation of *ZFP36L1* triggers an overexpression of the ZEB2 (zinc finger E-box binding homeobox 2) transcription factor, which, in turn, promotes an epithelial–mesenchymal transition (EMT). In this study, the authors demonstrated that BRF1 directly bound the *ZEB2* 3′UTR and decreased its expression [192]. Paradoxically, an overexpression of *ZFP36L1* was described in HCC patients as compared with cirrhotic livers [193,194] (Figure 7).

### 7.3. P-Bodies in HCC

For many cancers, it was suggested that P-bodies contribute to the activation of pro-tumoral pathways (e.g., MAPK, RAS (GTPase-activating protein-binding protein 1), Wnt or AKT signaling), thereby promoting cancer cell proliferation and invasion, while hindering cell death [41]. In HCC, the role of P-bodies is currently unknown, but emerging evidence indicates that some P-body components are involved in hepatic carcinogenesis. First, DCP1A is overexpressed in hepatic tumors as compared with peritumoral tissue and is considered as an unfavorable prognostic marker in HCC patients [195] (Figure 7). The CCR4-NOT complex plays an important role in liver homeostasis. Interestingly, the upregulation of the histone deacetylases 1 and 3 (HDAC1/3) promotes the overexpression of CNOT1 in hepatic cancer cells [14]. Finally, the RBP IGF2BP1 (IGF2 mRNA-binding protein-1) destabilizes the liver-cancer-associated lncRNA HULC (Highly Upregulated in Liver Cancer) in a CNOT1-dependent manner [196] (Figure 7). On the other hand, CNOT7, which is highly expressed in patients with HCC, correlates with a poor prognosis and a low STAT1 expression. Similar data were obtained in hepatic cancers cells (i.e., HepG2) and, conversely, the CNOT7 deficiency decreases the TGFβ level and NFκB signaling and induces STAT1 expression. Moreover, the supernatants of CNOT7-deficient HepG2 cells triggers IFNγ (interferon gamma) expression in NK (natural killer) cells (NK-92MI cells). Finally, co-culture experiments demonstrated that the sensitivity of HepG2 cells to NK-cell-dependent apoptosis is increased by CNOT7 deficiency [197]. This effect has been associated with an increased expression of CD107A (a marker of cytotoxic activity) at the surface of NK cells [197].

Together, these findings reveal that the role of CCR4-NOT and P-bodies in HCC remains totally unclear, and thus, intense efforts are still required to generate conclusions about their tumor-promoting or tumor-suppressive functions.

## 8. Conclusions

TTP family members and P-bodies are tightly involved in the development of chronic liver diseases and their progression toward HCC. These findings suggest that targeting TTP family members or P-bodies may represent a novel therapeutic approach for these disorders. However, the precise functions of these proteins remain to be clarified in the context of MASLD and ALD. TTP family members are still considered as “undruggable proteins”. To date, only unspecific molecules, such as metformin [169], demethylating agent (e.g., 5 Aza-2′deoxycytidine), MK2 inhibitors [73] or some Histone Deacetylase (HDAC) inhibitors (e.g., trichostatin-A) can modify the TTP expression/activity. Targeting miRNAs that regulate the expressions of TTP, BRF1 or BRF2 (Table 1) may also represent a potential therapeutic approach. More recently, some studies pointed out the benefits of some CCR4-NOT inhibitors against hepatic steatosis. Further investigations are still required to develop such molecules and to characterize their therapeutic potential. Taken together, current studies suggest that inhibiting P-bodies assembly may represent an appealing approach in the context of HCC, but such an approach should be carefully evaluated given that P-bodies assembly is also a protective mechanism in other contexts (e.g., HCV infection). Finally, identifying the mRNA targets of TTP family members and of P-bodies involved in the development of chronic liver diseases and HCC is also of high importance and may unravel new therapeutic targets. Finally, the TTP-dependent regulation of gene expression also involves other cytoplasmic compartments, namely, “Stress Granules”, which importantly control mRNA translation. Emerging evidence suggests that SGs importantly contribute to the development of MASLD and HCC. Deciphering their role may also open new therapeutic avenues.

## Figures and Tables

**Figure 1 cancers-17-00348-f001:**
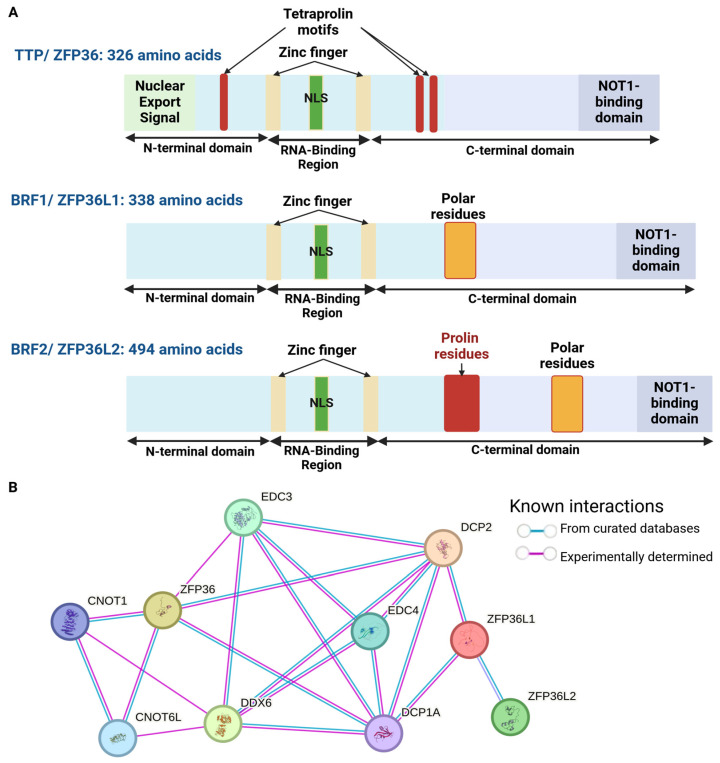
TTP structure, interaction and expression. (**A**) Protein domains of TTP, BRF1 and BRF2; TTP: Tristetraprolin; ZFP36: Zinc Finger Protein 36; BRF1: Butyrate response factor 1; BRF2: Butyrate response factor 2; ZFP36L1/2: Zinc Finger Protein 36 like 1/2. (**B**) TTP protein interaction. The blue lanes represent the interactions defined by databases, and the pink lanes represent experimentally determined interactions. CNOT1: CCR4-NOT Transcription Complex Subunit 1, CNOT6L: CCR4-NOT Transcription Complex Subunit 6 Like; DDX6: DEAD Box Helicase 6; DCP1A: Decapping mRNA 1 A; DCP2: Decapping mRNA 2; EDC3: Enhancer of mRNA Decapping 3; EDC4: Enhancer of mRNA Decapping 4; NLS: Nuclear Localization Sequence; ZFP36: Zing Finger Protein 36 homolog ZFP36L1: Zing Finger Protein C3H Type-Like 1; ZFP36L2: Zing Finger Protein C3H Type-Like 2.

**Figure 2 cancers-17-00348-f002:**
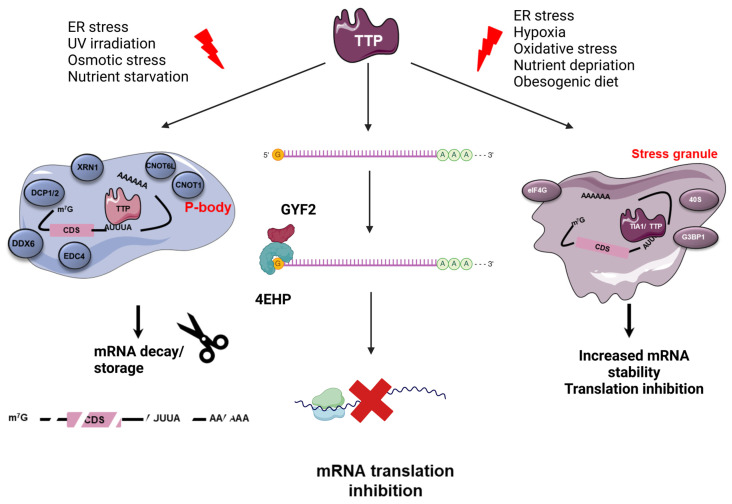
TTP-mediated mRNA regulation. TTP can affect the fate of mRNA in different ways. First, the assembly of P-bodies allows for mRNA decay thanks to the presence of several enzymes that promote the deadenylation, decapping and degradation of mRNA. In stress conditions, TTP also localizes in stress granules, where the mRNAs are kept translationally silent. Finally, TTP is involved in mRNA translation inhibition, notably by inhibition of the fixation of the complex elF4F for mRNA translation by recruiting the 4EHP-GYF2 complex. ER: Endoplasmic reticulum; UV: ultraviolet; GYF2: GRB10 interacting GYF protein 2; 4EHP: elF4E homologous protein; TTP: Tristetraprolin; CNOT1: CCR4-NOT Transcription Complex Subunit 1, CNOT6L: CCR4-NOT Transcription Complex Subunit 6 Like; DDX6: DEAD Box Helicase 6; DCP1A: Decapping mRNA 1 A; DCP2: Decapping mRNA 2; EDC4: Enhancer od mRNA Decapping 4; XRN1: 5′-3′ exoribonuclease 1; TIA1: T-cell intracellular antigen-1; G3BP1: Ras GTPase-activating protein-binding protein 1; elf4G: Eukaryotic translation initiation factor 4 gamma 1.

**Figure 3 cancers-17-00348-f003:**
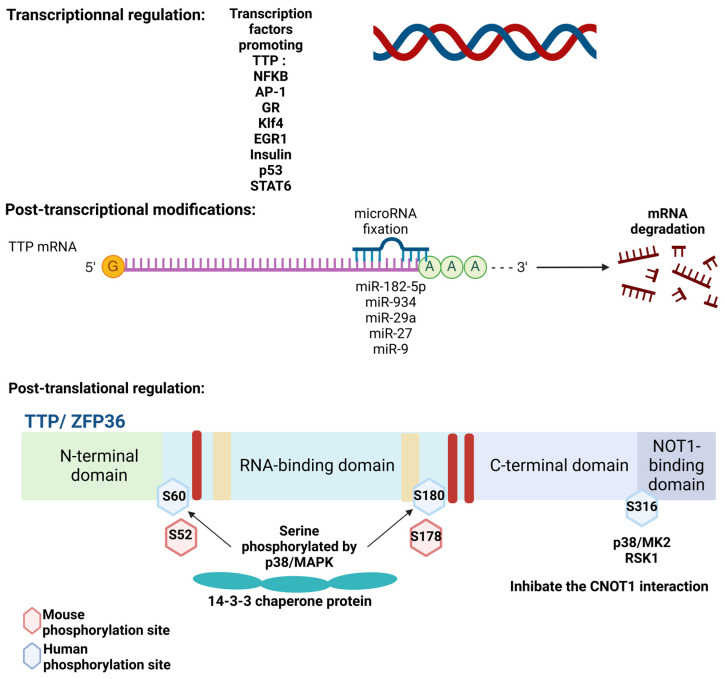
TTP family member regulation. TTP can be modulated at the transcriptional level by some factors, such as NFKB, AP-1, GR, KLF4 or EGR1, increasing the TTP expression, and CREB and STAT6, decreasing the TTP expression. At the post-transcriptional level, TTP mRNA can be targeted by some microRNA. TTP phosphorylation importantly regulates its activity (e.g., sequestration by 14-3-3 following MK2-dependent phosphorylation). TTP: Tristetraprolin; NFκB: Nuclear Factor kappa B; AP-1: Activating Protein 1; GR: Glucocorticoid Receptor; Klf4: Kruppel-like factor 4; EGR1: Early Growth Response 1; STAT6: Signal Transducer and Activator and Transcription 6; MAPK: Mitogen activated Protein Kinase; CNOT1: CCR4-NOT Transcription Complex Subunit 1; CREB: C-AMP Response Element-binding.

**Figure 4 cancers-17-00348-f004:**
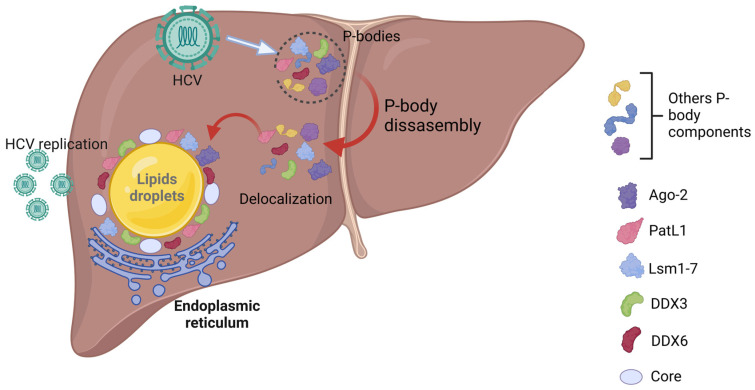
P-bodies in HCV infection. P-body disassembly and delocalization of P-body components in lipid droplets contribute to HCV replication. HCV: hepatitis C virus; P-body: processing body; Ago-2: Argonaute RISC Catalytic Component 2; PatL1: Pat1 like 1; Lsm1–7: Sm-Like protein 1–7; DDX3: DEAD-box helicase 3 X-linked; DDX6: DEAD box helicase 6.

**Figure 5 cancers-17-00348-f005:**
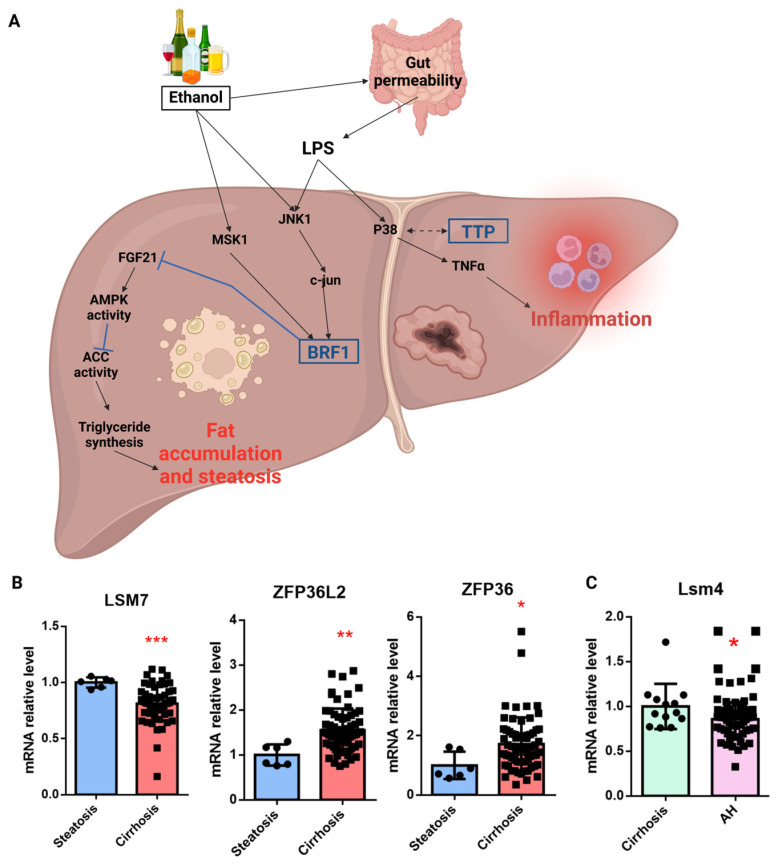
(**A**) Role of Tristetraprolin family members in alcoholic liver disease. Black arrows: activation of the pathways. Blue arrows: pathway inhibition. LPS: lipopolysaccharides; AMPK: AMP-activated protein kinase; MSK1: Mitogen and Stress Activated Protein Kinase 1; FGF21: fibroblast growth factor 21; ACC: acetyl-coA carboxylase; BRF1: Butyrate response factor 1; JNK1: c-jun-N-terminal kinase 1; TTP: Tristetraprolin; TNFα: tumor necrosis factor α. Created with Biorender.com. (**B**) Analysis of the Geodata set of patients (GSE103580) comparing steatosis with cirrhosis and (**C**) alcohol hepatitis with cirrhosis. All analysis was performed with RStudio (RStudio version 4.2.3 (2023-03-15 ucrt)) and Prism (version 10.4.1). Data were retrieved in September 2024. * *p* < 0.05; ** *p* < 0.01; *** *p* < 0.001.

**Figure 6 cancers-17-00348-f006:**
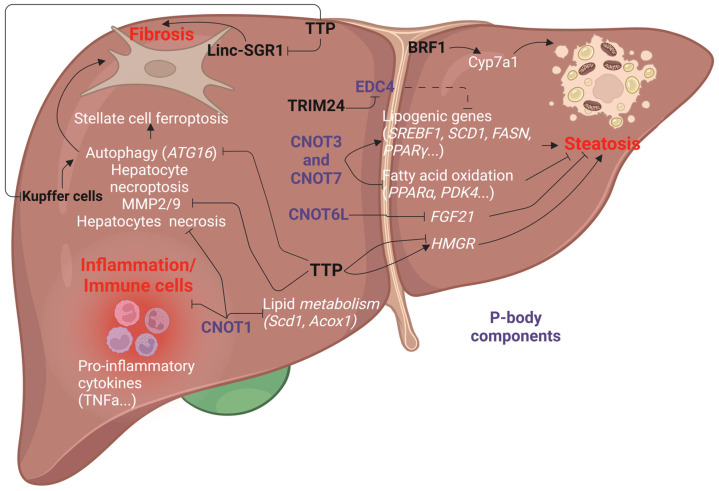
Role of TTP in MASLD. TTP promotes hepatic steatosis development by targeting FGF21, a hepatocyte involved in lipid metabolism, and HMGR, which is involved in cholesterol biosynthesis and liver gluconeogenesis. FGF21 is also a target of CNOT6L, a protein involved in processing bodies. CNOT7 promotes lipid uptake and synthesis. In MASH/fibrosis, TTP exerts an anti-inflammatory function in Kupffer cells. In HSCs, TTP exerts a dual function by (i) inhibiting TGFβ-induced HSCs activation and (ii) inhibiting HSC ferroptosis. CNOT1: CCR4-NOT Transcription complex subunit 1; CNOT3: CCR4-NOT Transcription complex subunit 3; CNOT6L: CCR4-NOT Transcription complex subunit 6 Like; CNOT7: CCR4-NOT Transcription complex subunit 7; TTP: Tristetraptolin; ATG16: Autophagy related 16 Like 1; MMP2/9: Matrix Metalloproteinase 2/9; TNFα: Tumor Necrosis Factor α; FGF21: Fibroblast Growth Factor 21; HMGR: 3-hydroxy-3-methylglutaryl-CoA-Reductase; FASN: Fatty Acid Synthase; ACACA: Acetyl-CoA Carboxylase alpha Protein; SCD1: Stearoyl-CoA desaturase 1; SREBP-1: Sterol Regulatory Element-Binding Protein 1; PPARγ: Peroxisome Proliferator Activated Receptor Gamma; PPARα: Peroxisome Proliferator Activated Receptor alpha; PDK4: Pyruvate Dehydrogenase Kinase 4; CPT1B: Carnitine Palmitoyltransferase 1B; TRIM24: Tripartite Motif Containing 24; EDC4: Enhancer Of mRNA Decapping 4; Elovl6: ELOVL Fatty Acid Elongase 6; HSD17B6: Hydroxysteroid 17 Beta Dehydrogenase 6; Cyp7a1: cholesterol 7α-hydroxylase; Scd1: Stearoyl Coenzyme A desaturase 1; Acox1: Acyl-CoA oxidase 1.

**Figure 7 cancers-17-00348-f007:**
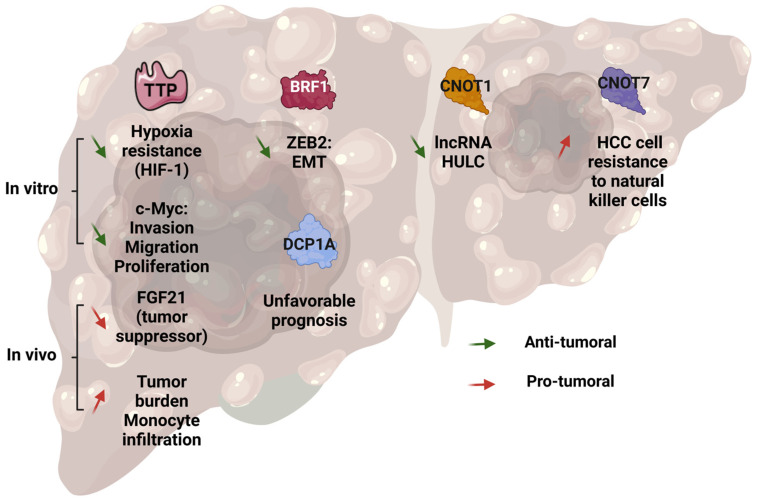
Role of TTP family members and P-bodies in hepatic carcinogenesis. TTP is involved in HCC development, as evidenced in liver-specific TTP KO mice, where a decrease of hepatic carcinogenesis was observed. But at a molecular level, it seems to have a protective effect by promoting apoptosis; decreasing cell proliferation; and providing invasion, migration and hypoxia resistance. The other members of the TTP family are also involved in HCC. BRF1 (ZFP36L1) decreases the epithelial mesenchymal transition (EMT), and BRF2 (5ZFP36L2) leads to an increase in neoplasm recurrence, tumor growth, metastasis and the modification of translation in cells. Some proteins known to be implicated in processing bodies, such as CNOT1, were studied in HCC. CNOT1 promotes apoptosis and decreases the expression of some pro-tumor lncRNA. TTP: Tristetraprolin; HIF-1: hypoxia inducible factor 1; c-Myc: cellular myelocytomatosis oncogene; BRF1: Butyrate response factor 1; BRF2: Butyrate response factor 1; CNOT1: CCR4-NOT transcription complex subunit 1; CNOT7: CCR4-NOT transcription complex subunit 7; HULC: Highly Upregulated in Liver Cancer; HHC: hepatocellular carcinoma; DCP1A: mRNA decapping enzyme 1A.

**Table 1 cancers-17-00348-t001:** miRs regulating ZFP36, ZFP36L1 and ZFP36L2 and their known functions in the liver.

MicroRNA	Model	References	In Liver Physiopathology?	References
ZFP36				
miR-9	Mouse primary neural stem cells + P19 cell line (epithelial)	[78]	n.a	n.a
miR-27	Myeloid Elav1 deleted mice + primary cell culture + HEK293T cell line	[62]	Fibrosis	[79]
miR-29a	Breast cancer cell lines + normal breast cells Pancreatic cancer + peritumoral tissue from patients	[61,80]	Fibrosis	[81,82]
miR-29c	Blood and urine from diabetic patients	[83]	n.a	n.a
miR-130b	Myeloma cell lines + primary human myeloma cells	[84]	n.a	n.a
miR-182	Cell line + mice breast cancer model	[85]	n.a	n.a
miR-182-5p	Human hepatocyte cell lines + mice model	[58]	Parasitic infection by *Schistosoma japonicum* + liver cancer and regeneration	[58,86,87]
mir-346	Human primary cell culture (from synovial tissues of arthritic patients)	[88]	n.a	n.a
miR-513a-5p	Breast carcinoma tissues and cell lines	[89]	HCC + fatty liver disease	[90,91]
miR-551b	Myeloma cell lines + primary human myeloma cells	[84]	n.a	n.a
miR-934	Human gastric cancer + peritumoral tissues	[92]	Hepatic metastasis of colorectal cancer	[93]
miR-let-7a	Mice primary mesangial cells	[94]	n.a	n.a
ZFP36L1				
miR-93-3p	Human cell lines + mice model	[95]	n.a	n.a
miR-96	Mice	[96]	Ameliorate alcohol-associated liver injury and liver cancer	[97]
miR-124	Astrocytes culture	[98]	Liver cancer with sorafenib resistance	[99]
miR-124-3p	Human adenocarcinoma	[100]	n.a	n.a
miR-129-5p	Normal and cancer human glial cells	[101]	Liver cancer + fatty liver disease	[102,103]
miR-182-5p	Human nasopharyngeal carcinoma	[104]	Liver regeneration	[86,87]
miR-377-3p	Human long fibroblast cell lines + primary human and mouse cells	[105]	Liver cancer	[106]
ZFP36L2				
miR-375	Human pancreatic cancer tissues	[107]	NASH + HCC	[108,109]
miR-429	Gastric cancer cell lines + human tissues	[110]	NASH + liver regeneration	[111,112]
miR-520d-3p	Human gastric cancer + peritumoral tissues	[113]	HCC	[114]

n.a: none available.

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
