# Peer review of "Tristetraprolin Family Members and Processing Bodies: A Complex Regulatory Network Involved in Fatty Liver Disease, Viral Hepatitis and Hepatocellular Carcinoma"

_cancers, 2025, doi:10.3390/cancers17030348_

Round 1
Reviewer 1 Report
Comments and Suggestions for Authors
The manuscript entitled “Tristetraprolin family members and Processing-Bodies: A com-2 plex regulatory network involved in Fatty Liver Disease and 3 hepatocellular carcinoma” has many mistakes, authors need to rectify many portions.
· I think remove the full stop “.” from the title.
· Please check the sentence, Is it correct? "Cirrhosis (e.g., lipid/glucose metabolism, inflammation, fibrosis, liver regeneration, carcinogenic processes) [8,9]." appears incomplete.
· Take care of abbreviations e.g. "MASH" is used, but it is not explicitly defined.
· Check the consistency in References, especially when mentioning studies or findings (e.g., [3, 5], [13, 15, 16]).
· Check the verbs throughout the manuscript. Eg. In statement "increasing evidence indicate that RNA Binding Proteins (RBPs)..." seems to contain a subject-verb agreement error. Should it be corrected to "increasing evidence indicates"?
· Check the TTP family members introduced clearly before diving into their regulation?
· Check abbreviations (e.g., NFκB, MAPK, LPS, HCC).
· Please check the descriptions of signaling pathways (e.g., NFκB, p38 MAPK/MK2) accurate and consistent with the latest literature?
· Please explore the current understanding of BRF1 and BRF2 regulation in the liver? Should future research directions or hypotheses be discussed?
· check references (e.g., [28], [50], [101])?
Good Luck!
Comments on the Quality of English LanguageLanguage must be correct.
Author Response
Reviewer-1
Comment 1: The manuscript entitled “Tristetraprolin family members and Processing-Bodies: A com-2 plex regulatory network involved in Fatty Liver Disease and 3 hepatocellular carcinoma” has many mistakes, authors need to rectify many portions.
Response: We thank the reviewer for the constructive comments, which helped us to improve our manuscript.
Comment 2: I think remove the full stop “.” from the title.
Response: The full stop has been removed from the title.
Comment 3: Please check the sentence, Is it correct? "Cirrhosis (e.g., lipid/glucose metabolism, inflammation, fibrosis, liver regeneration, carcinogenic processes) [8,9]." appears incomplete.
Response: The sentence is the following: “Epigenetic defects (e.g., DNA methylation, histones variants and modifications) of gene expression importantly contribute to the alteration of several processes involved in FLD development (e.g., lipid/glucose metabolism, inflammation, fibrosis, liver regeneration, carcinogenic processes) [8,9]. The sentence is correct. We carefully check if incomplete sentences are present in the manuscript.
Comment 4: Take care of abbreviations e.g. "MASH" is used, but it is not explicitly defined.
Response: We apologize for the lack of clarity. This issue has been corrected.
Comment 5: Check the consistency in References, especially when mentioning studies or findings (e.g., [3, 5], [13, 15, 16]).
Response: The references 3, 5 were correctly mentioned in the manuscript and are supporting the sentence: “Then, the accumulation of other damages, including oxidative stress, lipotoxicity, mitochondrial dysfunction and endoplasmic reticulum stress, triggers hepatocytes death and the activation of hepatic stellate cells (HSCs), which in turn foster hepatic fibrosis and the progressive loss of hepatic functions [3,5].”
The references 13, 15 and 16 are also consistent and correspond to other reviews discussing in details the regulatory mechanisms of AUBPs: “While some AUBPs stabilize mRNAs (e.g., HuR), others promote mRNA decay, such as tristetraprolin family members (i.e., tristetraprolin, TTP; Butyrate Response Factor 1, BRF1 and Butyrate Response Factor 2, BRF2) [13, 15, 16]”
Comment 6: Check the verbs throughout the manuscript. Eg. In statement "increasing evidence indicate that RNA Binding Proteins (RBPs)..." seems to contain a subject-verb agreement error. Should it be corrected to "increasing evidence indicates"?
Response: We carefully checked all the manuscript, and we have corrected all subject-verb errors. We thank the reviewer for noticing this issue.
Comment 7: Check the TTP family members introduced clearly before diving into their regulation?
Response: At the beginning of the review, in the paragraph 2.1 “AUBPs at a glance” and 2.2 “TTP family members: from structure to function”, we are introducing TTP family members and these paragraphs are before the paragraph 3 “TTP family members regulation in the liver”
Comment 8: Check abbreviations (e.g., NFκB, MAPK, LPS, HCC).
Response: All abbreviations have been carefully checked and corrected when necessary.
Comment 9: Please check the descriptions of signaling pathways (e.g., NFκB, p38 MAPK/MK2) accurate and consistent with the latest literature?
Response:
For the P38/MK2 signaling, we are stating: “Among them, the p38 MAPK/MK2 plays a major role in TTP regulation. Indeed, the phosphorylation of TTP on Ser60 and Ser186 in human, or Ser52 and Ser178 in mouse by MK2, triggers TTP sequestration by the 14-3-3 chaperone in the cytosol [24,100]”. This statement is correct and supported by ref 24 and 100 but also PMID: 20594301, which has been added in our manuscript.
For NFkB, we discuss that NFkB triggers ZFP36 transcription and this statement is correct.
Finally, in the paragraph 2.4 “Other TTP-mediated effect”, we described the ability of TTP to prevent p65 nuclear translocation. This statement is supported by ref 46: “Schichl, Y.M.; Resch, U.; Hofer-Warbinek, R.; de Martin, R. Tristetraprolin impairs NF-kappaB/p65 nuclear translocation. J Biol Chem 2009, 284, 29571-29581, doi:10.1074/jbc.M109.031237.”
Comment 10: Please explore the current understanding of BRF1 and BRF2 regulation in the liver? Should future research directions or hypotheses be discussed?
Response: We agree with the reviewer that this is an important aspect to consider. Unfortunately, we are unable to answer this question, given the lack of knowledge for ZFP36L1 and ZFP36L2. Nevertheless, table 1 provides several examples of BRF1/2 regulation by miRNAs. Targeting these miRNAs may represent an indirect way to inhibit/restore BRF1/BRF2 expression. We have added one sentence in our conclusion to suggest this potential direction.
“Targeting miRNAs regulating the expression of TTP, BRF1 or BRF2 (Table 1) may also represent a potential therapeutic approach.”
Comment 11: check references (e.g., [28], [50], [101])?
Response:
In the paragraph 3.1 “TTP”, the reference 28 was indeed not the good one. We replaced the ref 28 by PMID: 23212617
The reference 50 refers to the regulation of TTP by insulin and is cited in the following sentence: “Besides these inflammatory signaling, TTP level is also stimulated by insulin signaling in adipocytes, thus highlighting its metabolic role [50,51] (Now [52,53]) (figure 4).”
The references 101, “Marchese, F.P.; Aubareda, A.; Tudor, C.; Saklatvala, J.; Clark, A.R.; Dean, J.L. MAPKAP kinase 2 blocks tristetraprolin-directed mRNA decay by inhibiting CAF1 deadenylase recruitment. J Biol Chem 2010, 285, 27590-27600, doi:10.1074/jbc.M110.136473.”, was cited in the following sentence “Moreover, this phosphorylation by MK2 inhibits the recruitment of CAF1 and thus impairs mRNA deadenylation, in a 14-3-3 independent manner [101] (Now [105]).”
Reviewer 2 Report
Comments and Suggestions for Authors
Review on the manuscript titled “Tristetraprolin family members and Processing-Bodies: A complex regulatory network involved in Fatty Liver Disease and hepatocellular carcinoma.”
The review addresses the impact of Tristetraprolin family across Liver diseases, in particular Fatty Liver disease (FLD), and hepatocellular carcinoma (HCC).
In the introduction, the authors outlined the etiology and dynamic of FLD and HCC, including their epidemiological status and therapeutic routines, as well as multiple genetic – associated specifics including epigenetic, genetic and other factors, such as miRNA/lncRNA impact.
Therein, the authors underline the specific mRNA-bound proteins (RBP), such as AUBPs (AU rich element binding proteins) capable of binding RNA at various specific regions (UTR, introns, etc) and apparently regulating mRNA processing pathway, while stabilizing mRNAs. Another RBP class target mRNA decay pathway (TTP, BRF1, BRF2). This passage outlines the objects/subjects of the review: AUBP family proteins, TTP in particular.
The involvement of TTP genes into mRNA degradation is explicitly described in the next chapters:
Chapter “2. TTP-dependent mRNA regulation” comprises several subchapters:
2.1. AUBPs at a glance.: the authors reported the specifics and pathways AUBPs are involved in the course of mRNA processing/editing.
2.2. TTP-family members: from structure to function.
This subchapter reports specifics of TTPs, majorly outlined in Fig.1a (3 TTP family members considered: TTP, BRF1, BRF2). Fig. 1b features string-db.org maintained gene network, but it’s not quite clear why no TTP members are present therein.
2.3. TTP-family members: a complex regulatory mechanism?
This chapter underlines a complexity of TTP-mediated genes network, and illustrates it in several issues therein.
2.4. TTP-mediated mRNA regulation.
This is the longest chapter featured with fig. 2 outlining the TTP pathways induced by stress.
Next, the authors present two last chapters with explicit description of TTP family specific functions in liver
3. TTP family members regulation in the liver.
Chapter contains following subchapters:
3.1. TTP.
Herein, the authors underline the vast regulation of ZFP genes with miRNA by compiling Table 2, but not for TTP. That’s a bit strange.
Also the chapter contains Fig. 3 containing several GPCR and kinase mediated pathways, for which TTPs are apparently involved.
3.2. BRF1 and BRF2.
This chapter majorly conveys that “The regulation of BRF1 and BRF2 is poorly documented and totally unknown in the liver”, but still the authors managed to compile some facts on their metabolism in liver.
Next follows chapter 4: Role of TTP family members and P-Bodies in viral hepatitis
It comprises following sections:
4.1. Hepatitis B virus.
4.2. Hepatitis C virus (Fig. 4)
Next are:
5. Role of TTP family members in Alcohol-Related Liver Diseases (Fig.5).
6. Role of ZFP36 family members and P-bodies in MASLD, including:
6.1. TTP family members in MASLD.
6.2. P-bodies in MASLD (Fig. 6)
7. Role of ZFP36 family members and P-bodies in HCC, including:
7.1. TTP and HCC.
7.2. BRF1 and BRF2 in HCC (Fig.7)
7.3. P-bodies in HCC (Fig.7)
8. Conclusions
Herein, the authors state that:
“TTP family members and P-bodies are tightly involved in the development of chronic liver diseases and their progression toward HCC. These findings suggest that targeting TTP family members or P-bodies may represent a novel therapeutic approach for these disorders.”
Overall, the authors did a profound work of literature compilation and trying to compile it in a sensible draft. While there could be an interest from researchers/clinicians in the field, the draft/manuscript is poorly positioned, since TTP family is quite non-specific housekeeping genes, affecting/regulating which would be unbearable.
The multiple instances of TTP genes involvement in the certain networks make the manuscript overstretched, also due to manuscript vague positioning and the hard to follow logics of what’s going on. Also, the concise linkage of TTP effect specifically to HCC and FLD is rather doubtful, since mRNA degradation is a typical housekeeping process. Some notes are presented below.
1) P-bodies: all housekeeping processes including transcription, splicing, histone modification, etc. occur in specific vesicle compartments for assuring corresponding processing factors/proteins concentration rate. mRNA degradation cascade is not an exclusion.
2) Abstract: “In this review, we discuss the role of this regulatory mechanism in Metabolic Dysfunction-Associated Steatotic Liver Disease (MASLD), Alcohol-related Liver Disease (ALD), hepatic viral infections and HCC.” A range of diseases aren’t mentioned in the title, should be rephrased either in the title or in the article body.
3) “Figure 1B: TTP protein interaction”. It is completely incomprehensive shot. Three TTP genes (TTP, considered in the article should be present in gene network. Major GO categories corresponding to the pathway should be outlined in the figure caption (see example below).
4) Table 1 is better off to outsource to Supplementary, since it’s not directly connected to TTP family considered in the chapter: Fig. 3 would be more relevant/enough.
5) Please, provide more transparent and concise logic within each chapter when using outsourcing genes, networks, etc.
6) I’ve built 2 networks based on the genes set the authors highlighted, for example:
1) Gene network recovered on 3 TTP genes used: TTP (ADAMTS13, BRF1, BRF2):

|
#color |
term ID |
term description |
observed gene count |
background gene count |
false discovery rate |
matching proteins in your network (labels) |
|
red |
GO:0006383 |
Transcription by RNA polymerase III |
8 |
33 |
7.90E-15 |
SNAPC1,SNAPC2,TBP,SNAPC4,SNAPC5,GTF3C4,SNAPC3,BRF1 |
|
blue |
GO:0009301 |
snRNA transcription |
5 |
14 |
4.88E-09 |
SNAPC1,SNAPC2,SNAPC4,SNAPC5,SNAPC3 |
|
yellow |
GO:0006352 |
DNA-templated transcription, initiation |
5 |
131 |
1.97E-05 |
BRF2,TBP,SNAPC5,GTF3C4,BRF1 |
|
limegreen |
GO:0061158 |
3-UTR-mediated mRNA destabilization |
3 |
17 |
0.00017 |
ZFP36L2,ZFP36L1,ZFP36 |
2) Network based on TTP seed only:

|
#color |
term ID |
term description |
observed gene count |
false discovery rate |
matching proteins in your network (labels) |
|
blue |
GO:0006956 |
Complement activation |
5 |
2.38E-06 |
C3,CFHR1,CFHR3,CFH,CFI |
|
red |
GO:0007596 |
Blood coagulation |
5 |
0.00013 |
VWF,DGKE,GP1BA,F8,ADAMTS13 |

Author Response
Reviewer-2
Review on the manuscript titled “Tristetraprolin family members and Processing-Bodies: A complex regulatory network involved in Fatty Liver Disease and hepatocellular carcinoma.”
The review addresses the impact of Tristetraprolin family across Liver diseases, in particular Fatty Liver disease (FLD), and hepatocellular carcinoma (HCC).
In the introduction, the authors outlined the etiology and dynamic of FLD and HCC, including their epidemiological status and therapeutic routines, as well as multiple genetic – associated specifics including epigenetic, genetic and other factors, such as miRNA/lncRNA impact. Therein, the authors underline the specific mRNA-bound proteins (RBP), such as AUBPs (AU rich element binding proteins) capable of binding RNA at various specific regions (UTR, introns, etc) and apparently regulating mRNA processing pathway, while stabilizing mRNAs. Another RBP class target mRNA decay pathway (TTP, BRF1, BRF2). This passage outlines the objects/subjects of the review: AUBP family proteins, TTP in particular. The involvement of TTP genes into mRNA degradation is explicitly described in the next chapters:
We thank the reviewer for the constructive comments, which help us to improve our manuscript.
Specific Comments:
Comment 1: Paragraph 2.2. TTP-family members: from structure to function.
This subchapter reports specifics of TTPs, majorly outlined in Fig.1a (3 TTP family members considered: TTP, BRF1, BRF2). Fig. 1b features string-db.org maintained gene network, but it’s not quite clear why no TTP members are present therein.
Response: In the figure 1B, the network shows direct interactions between TTP (ZFP36) and CNOT1, CNOT6L, DCP2, DCP1A and EDC3. We have modified the network by adding ZFP36L1 and ZFP36L2 (BRF1 and BRF2). We also modified some settings of our STRING network (interaction sources: experiments and database; meaning of network edges: evidence; medium confidence 0.400). We thank the reviewer for noticing this issue.
Comment 2: Paragraph 3.1. TTP.
Herein, the authors underline the vast regulation of ZFP genes with miRNA by compiling Table 2, but not for TTP. That’s a bit strange. Also the chapter contains Fig. 3 containing several GPCR and kinase mediated pathways, for which TTPs are apparently involved.
Response: We disagree with the reviewer. ZFP36 corresponds to the name of the gene for TTP. The table 1 gathers 12 miRNAs regulating ZFP36 (TTP).
Comment 3: “TTP family members and P-bodies are tightly involved in the development of chronic liver diseases and their progression toward HCC. These findings suggest that targeting TTP family members or P-bodies may represent a novel therapeutic approach for these disorders.”
Overall, the authors did a profound work of literature compilation and trying to compile it in a sensible draft. While there could be an interest from researchers/clinicians in the field, the draft/manuscript is poorly positioned, since TTP family is quite non-specific housekeeping genes, affecting/regulating which would be unbearable.
Response: We disagree with this comment. TTP is not a housekeeping gene but an inducible early-response gene and a key regulator of inflammation and stress response. Alteration of its expression/activity has been involved in a wide range of inflammatory diseases and cancers. 926 publications on TTP are currently available and among the latest, the therapeutic potential of TTP targeting has been highlighted. Please see the following examples:
Roles of Tristetraprolin in Tumorigenesis. Park JM, Lee TH, Kang TH.Int J Mol Sci. 2018 Oct 29;19(11):3384. doi: 10.3390/ijms19113384.PMID: 30380668
Clinical implications of tristetraprolin (TTP) modulation in the treatment of inflammatory diseases. Snyder BL, Blackshear PJ.Pharmacol Ther. 2022 Nov;239:108198. doi: 10.1016/j.pharmthera.2022.108198. Epub 2022 May 5.PMID: 35525391
Tristetraprolin as a Therapeutic Target in Inflammatory Disease. Patial S, Blackshear PJ.Trends Pharmacol Sci. 2016 Oct;37(10):811-821. doi: 10.1016/j.tips.2016.07.002. Epub 2016 Aug 5.PMID: 27503556
Therefore, we believe that our manuscript is well positioned in the field and provide an extensive review of the role of TTP in chronic liver diseases and HCC, which remains poorly known and of high interest for people working in hepatology but also in other diseases on these regulatory mechanisms. We would like to stress also the fact that they are currently no review on TTP/P-Bodies in the context of viral hepatitis, Alcohol-related hepatitis etc. This reinforces the novelty of our review.
The multiple instances of TTP genes involvement in the certain networks make the manuscript overstretched, also due to manuscript vague positioning and the hard to follow logics of what’s going on. Also, the concise linkage of TTP effect specifically to HCC and FLD is rather doubtful, since mRNA degradation is a typical housekeeping process. Some notes are presented below.
- Comment 4: P-bodies: all housekeeping processes including transcription, splicing, histone modification, etc. occur in specific vesicle compartments for assuring corresponding processing factors/proteins concentration rate. mRNA degradation cascade is not an exclusion.
Response: We agree with the reviewer that P-Bodies assembly is a dynamic process involved in RNA decay and thus several physiological processes. However, like for histone modifications, RNA splicing etc, a disequilibrium of this process, mediated by abnormal expression of TTP family members or P-Bodies components (EDC4, CNOT6L), not only alters P-Bodies assembly but contribute also to pathological situations. Discussing these alterations is of high interest for human diseases, in particular in the context of liver diseases and HCC, where the role of TTP family members and of P-Bodies remains poorly characterized. Deciphering these mechanisms may also unravel novel therapeutic approaches, as suggested for CNOT6L inhibitors in the context of MASLD (discussed in the conclusion of our manuscript).
- Comment 5: Abstract: “In this review, we discuss the role of this regulatory mechanism in Metabolic Dysfunction-Associated Steatotic Liver Disease (MASLD), Alcohol-related Liver Disease (ALD), hepatic viral infections and HCC.” A range of diseases aren’t mentioned in the title, should be rephrased either in the title or in the article body.
Response: “Fatty Liver disease” gathers MASLD and Alcohol-related Liver Disease. We agree that this does not include viral hepatitis. We have modified the title of our manuscript as follow: “Tristetraprolin family members and Processing-Bodies: A complex regulatory network involved in Fatty Liver Disease, viral Hepatitis and hepatocellular carcinoma”
- Comment 6 : “Figure 1B: TTP protein interaction”. It is completely incomprehensive shot. Three TTP genes (TTP, considered in the article should be present in gene network. Major GO categories corresponding to the pathway should be outlined in the figure caption (see example below).
Response: The goal of the network in 1B is to highlight the interactions between P-Bodies components and TTP family members (ZFP36, ZFP36L1 and ZFP36L2). Please note that ZFP36, ZFP36L1 and ZFP36L2 correspond to the gene names of TTP, BRF1 and BRF2. We have modified the network to incorporate ZFP36L1 and ZFP36L2 as suggested by the reviewer.
- Comment 7: Table 1 is better off to outsource to Supplementary, since it’s not directly connected to TTP family considered in the chapter: Fig. 3 would be more relevant/enough.
Response: We disagree with the reviewer. The table 1 clearly provides examples of miRNAs regulating TTP family members (i.e., ZFP36, ZFP36L1 and ZFP36L2).
-Comment 8: Please, provide more transparent and concise logic within each chapter when using outsourcing genes, networks, etc.
Response: The comment of the reviewer is confusing. We believe that the plan of our review is quite clear, with many paragraphs on specific topics (HBV, HCV, MASLD, ARLD, HCC…), which allow the readership to focus on their specific interest.
-Comment 9: I’ve built 2 networks based on the genes set the authors highlighted, for example: Gene network recovered on 3 TTP genes used: TTP (ADAMTS13, BRF1, BRF2):
Response: We are not sure to understand the comment raised by the reviewer. Moreover, there is confusion in the provided network between BRF1/2 (Butyrate Response Factor-1/2, alias ZFP36L1/L2) and BRF1/2 (General Transcription Factor IIIB Subunit). They are not the same proteins.
Reviewer 3 Report
Comments and Suggestions for Authors
Dear editor, dear authors,
1. Could you elaborate on the specific mechanisms by which TTP family members interact with other RNA-binding proteins or non-coding RNAs to regulate mRNA stability and translation in the context of fatty liver disease? Are there particular interactions that have been identified as critical for the progression of the disease?"
2. The manuscript discusses the role of P-bodies in various liver diseases. Could you provide more details on how the composition and function of P-bodies differ between fatty liver disease and viral hepatitis, and how these differences might influence the effectiveness of potential therapeutic interventions targeting P-bodies?"
3. While the manuscript highlights the potential of targeting TTP family members and P-bodies for therapeutic purposes, could you discuss the current limitations in developing effective drugs to modulate these pathways? Additionally, are there any emerging strategies or technologies that could overcome these challenges and enhance the therapeutic potential of this approach?"
Author Response
Reviewer-3
The manuscript examines the role of tristetraprolin (TTP) family members and processing-bodies in fatty liver disease and hepatocellular carcinoma, providing insights into their regulation of gene expression. It highlights potential therapeutic targets to prevent disease progression. However, the study's complexity may limit general applicability. Further mechanistic studies and in vivo models could enhance its contributions. Here are three detailed questions based on the content of the manuscript:
We thank the reviewer for his/her time to review our manuscript and for the constructive comments, which allowed us to improve our review.
Comment 1: Could you elaborate on the specific mechanisms by which TTP family members interact with other RNA-binding proteins or non-coding RNAs to regulate mRNA stability and translation in the context of fatty liver disease? Are there particular interactions that have been identified as critical for the progression of the disease?"
Response: This is a very interesting comment. Indeed, several interplay between RBPs and non-coding RNAs have been described. This aspect has been extensively described in a previous review:
“MicroRNAs, Tristetraprolin Family Members and HuR: A Complex Interplay Controlling Cancer-Related Processes. PMID: 35884580 PMCID: PMC9319505 DOI: 10.3390/cancers14143516”
Unfortunately, the interplay between TTP and ncRNA is currently unknown in the liver. In the table 1, we have provided a list of potential miRNAs regulating TTP expression and deregulated in the context of liver diseases.
Comment 2: The manuscript discusses the role of P-bodies in various liver diseases. Could you provide more details on how the composition and function of P-bodies differ between fatty liver disease and viral hepatitis, and how these differences might influence the effectiveness of potential therapeutic interventions targeting P-bodies?"
Response: This is also a very interesting question. Unfortunately, we are unable to answer this comment for HBV/HCV, given the low number of available studies on this topic. For HCV, current data suggest that P-Bodies are protective against HCV, while in FLD, P-Bodies may favor hepatic steatosis. Together, these findings suggest cautions regarding the targeting of P-Bodies (for instance with CNOT6L inhibitors) in HCV-infected patients. A small sentence has been added in the conclusion.
“Taken together, current studies suggest that inhibiting P-Bodies assembly may represent an appealing approach in the context of HCC but such approach should be carefully evaluated, given that P-Bodies assembly is also a protective mechanism in other contexts (e.g., HCV infection).”
Comment 3: While the manuscript highlights the potential of targeting TTP family members and P-bodies for therapeutic purposes, could you discuss the current limitations in developing effective drugs to modulate these pathways? Additionally, are there any emerging strategies or technologies that could overcome these challenges and enhance the therapeutic potential of this approach?"
Response: We agree with the reviewer that this is an important concern in the field. Currently, TTP family members are considered as “undruggable” and only unspecific approach can be used to inhibit/increase TTP expression (e.g., HDAC inhibitors, DNA demethylating agents, microRNAs mimics). Another possibility would be to target downstream mechanisms, such as TTP and emerging studies indicate that CCR4/NOT inhibitors, such as CNOT6L inhibitors may provide novel therapeutic options, as evidenced in MASLD. These aspects were discussed in the conclusion of our manuscript:
“TTP family members are still considered as “undruggable proteins”. To date, only unspecific molecules, such as metformin [165], demethylating agent (e.g., 5 Aza-2’deoxycytidine), MK2 inhibitors [106], or some Histone Deacetylase (HDAC) inhibitors (e.g., trichostatin-A) can modify TTP expression/activity. More recently, some studies have pointed out the benefit of some CCR4-NOT inhibitors on hepatic steatosis”
Round 2
Reviewer 2 Report
Comments and Suggestions for Authors
Please See the attachment below

Author Response
Comment 1: The authors have essentially addressed my comments.
I’m sorry to mix the issues since BRF1, BRF2 are rather ambiguous names and are
employed in distinct processes: Transcription processing vs P-bodies composition, e.g.
correspond to different genes. The authors should mention all gene aliases of TTP family
in figure legends/introduction of TTP family names: TTP (ZFP36), BRF1 (ZFP36L1),
BRF2 (ZFP36L2) to avoid any ambiguity.
Response: The correspondences between TTP/ZFP36, BRF1/ZFP36L1 and BRF2/ZFP36L2 are indicated in the paragraph 2.2 "TTP family members: from structure to function".
Comment 2: Besides, I suggest my version of Fig.1b for clarity listed below.(same genes set as in
original Fig.1b). It illustrates P-body and TTP family intercalation more vividly.
Response: We appreciate the effort of the reviewer to help us in the design of the figure 1B. However, our initial goal was to highlight the protein-protein interactions, rather than the Gene Ontology processes. Moreover, the enriched processes provided do not bring new data to the figure, as it is well established that the different proteins of the network belong to P-Bodies.